# Dynamic Task Scheduling Scheme for Processing Real-Time Stream Data in Storm Environments

Dojin Choi [1], Hyeonwook Jeon [1], Jongtae Lim [1], Kyoungsoo Bok [2] and Jaesoo Yoo [1],*

1   Department of Information and Communication Engineering, Chungbuk National University, Chungdae-ro 1, Seowon-Gu, Cheongju 28644, Korea; mycdj91@chungbuk.ac.kr (D.C.); wooky@chungbuk.ac.kr (H.J.); jtlim@chungbuk.ac.kr (J.L.)
2   Department of SW Convergence Technology, Wonkwang University, Iksandae 460, Iksan 54538, Korea; ksbok@wku.ac.kr
*   Correspondence: yjs@cbnu.ac.kr or yjs@chungbuk.ac.kr; Tel.: +82-43-261-3230

**Abstract:** Owing to the recent advancements in Internet of Things technology, social media, and mobile devices, real-time stream balancing processing systems are commonly used to process vast amounts of data generated in various media. In this paper, we propose a dynamic task scheduling scheme considering task deadlines and node resources. The proposed scheme performs dynamic scheduling using a heterogeneous cluster consisting of various nodes with different performances. Additionally, the loads of the nodes considering the task deadlines are balanced by different task scheduling based on three defined load types. Based on diverse performance evaluations it is shown that the proposed scheme outperforms the conventional schemes.

**Keywords:** dynamic task scheduling; stream processing; Storm; deadline; load balancing





## 1. Introduction

With the increase in the use of various media, such as social media and the Internet of Things, large amounts of real-time stream data are generated in various forms. In a sensor network, real-time stream data are used to detect equipment malfunction and monitor mal-operation [1,2]. A system that identifies equipment malfunction or maloperation is unable to serve its objective if it fails to assess scenarios and provide appropriate notifications, and therefore, real-time big data processing technology is required for responding within the time defined by users [3,4]. Stream processing platforms are commonly used for real-time stream processing of big data, such as Spark, Storm, and Flink [5–7]. These platforms consistently perform stream processing by operating Java Virtual Machines (JVM) once, thus solving the problem of re-operating JVMs in Hadoop [8,9]. Moreover, in order to solve the response problem, intermediate results of the processing are maintained in the memory for minimizing disk I/O. In particular, the main objective of Storm is real-time stream processing, different from Spark and Flink, which aim at both stream and batch processing. Therefore, this study focuses on Storm, which can efficiently perform stream processing of big data. Storm has the advantages of high data processing speed, scalability, fault tolerance, reliability, and low operation difficulty [7]. In Storm, Nimbus, which is the master node, delivers the data to the supervisor of a worker node, which is a slave, for stream processing data input in real time. During this process, tasks are distributed by round-robin fashion. However, the input tasks may have varying degrees of computational complexity, which may apply loads on a particular worker node. A delay may occur in real-time processing if the loads on the worker nodes are not considered.

Various studies have been conducted for resolving this problem [10–21]. In some study, simply calculating the traffic cannot solve the problem of load imbalance on worker nodes. In addition, only an environment in which the expected task processing time does not fluctuate significantly was considered by including simple queries, such as data

search. Therefore, scenarios in which complex operation queries are requested need to be considered in addition to simple search queries. Some studies cannot deal with real-time tasks given a specific deadline.

This study proposes a dynamic task scheduling scheme in which task deadlines and the node resources in Storm are considered. The proposed scheme manages the states of the worker nodes to assess the cause of the loads on them and performs dynamic scheduling based on the load type. For performing dynamic scheduling, the priority of task is determined based on the deadline information and their processing costs. For load balancing, tasks with higher priorities are redistributed over the relatively lower loaded nodes. For performing dynamic task scheduling in heterogeneous cluster environments with different performance levels, the performance of each worker node is recorded, and tasks with higher priorities are assigned to the worker nodes with better performance, to complete tasks within the given deadline. Consequently, the proposed method distributes loads on the worker nodes to prevent bottleneck in real-time processing. This study has the following contributions:

1.  Dynamic scheduling based on the cluster status: Cluster worker nodes may have varying performance depending on the operation scenario. Each worker node has a different task throughput; thus, a worker node with a higher throughput must process a larger number of tasks. In this study, dynamic scheduling for task distribution is performed by considering the diversity in the performance of cluster worker nodes.
2.  Dynamic scheduling considering the load type: The load types of a worker node are classified in this study as CPU, memory, or CPU and memory loads based on the number of resources used. The number of resources used per task is monitored, and dynamic scheduling is performed to redistribute the tasks based on the defined load types.
3.  Dynamic scheduling considering the task deadline: If a task is not processed within a deadline during real-time stream processing, the processing result becomes insignificant. In this study, dynamic scheduling is performed by considering the deadline for each task based on the loads generated during the stream processing.

The remainder of this paper is organized as follows. Section 2 explains the characteristics and problems of conventional scheduling schemes. Section 3 describes the dynamic task scheduling proposed in this paper. In Section 4, the excellence of the proposed scheme is proved by presenting the comparison of the performances of conventional schemes and the stream processing scheme proposed in this paper. The conclusions and directions for future research are provided in Section 5.

## 2. Related Work

The round-robin scheduler provided by default in Storm has the advantage of simple implementation; however, it causes an excessive number of tasks to be allocated to low-performance worker nodes because the performance of the worker nodes and a distributed environment with various performance are not considered. In this case, response delay and task loss occur during real-time stream processing of data, which causes failure in task completion within the deadline. For resolving these issues, various scheduling schemes have been examined in which the loads on the worker nodes in a real-time stream environment are considered [10–21].

A study [10] proposed a task scheduling scheme for distributing tasks by predicting the processing times of the tasks based on the performance of the worker nodes for rapid search performance. The tasks that require longer processing time than expected are redistributed to standardize the processing times of the worker nodes. However, the study aimed to perform relatively simple tasks, such as search, and thus, an environment in which complex tasks, such as the k-means algorithm, are simultaneously input must be considered.

T-Storm is a task scheduling scheme in which the traffic between the worker nodes and between the processes is considered [11]. The data on the CPU usage and traffic between

the worker nodes and between the processes measured by load monitors are stored in a database. Subsequently, a scheduler in the master sequentially distributes the tasks starting with those with the lowest traffic between the worker nodes and between the processes based on the data stored in the database.

R-Storm is a task scheduling scheme in which the resources of the worker nodes and the expected processing costs of the tasks are considered [12]. R-Storm distributes the tasks to the most suitable worker nodes by comparing the CPU usage, memory, and network costs of the worker nodes with the corresponding expected values to be used by the tasks. After presenting the expected usage of the resources by the tasks and the free resources (CPU, memory, and network cost) of the worker nodes as three-dimensional dots, each task is allocated to the worker nodes, starting with the closest node based on the three-dimensional distance. Accordingly, the task processing efficiency is improved by selecting the most appropriate worker node for the expected processing costs.

A study [13] proposed static and dynamic task scheduling schemes based on the CPU usage and network bandwidths of the worker nodes. The default task allocation in [13] was static allocation, which was based on the network bandwidth and the transmission amount of a tuple. Subsequently, dynamic scheduling is proceeded at certain intervals (5 min). Dynamic scheduling identifies load occurrence when the CPU usage of a worker node exceeds the threshold. Subsequently, the task at the loaded worker node is migrated to a worker node that is unloaded by considering the CPU usage and the network costs. Accordingly, dynamic scheduling is performed based on the loads on the worker nodes. However, important resources for processing tasks, such as memory usage, were not considered, and dynamic scheduling also fails to consider the task deadline; therefore, it is incapable of processing urgent tasks.

Studies [14,15] proposed a scheduling scheme which includes the topology of Storm in addition to network traffic and worker nodes' resources which were considered in T-Storm and R-Storm. A study [14] proposed a task scheduling scheme based on the topology and resources for a heterogeneous cluster environment. Based on the topology information of Storm, the amounts of communications within a group and between groups are defined to be used for task allocation. Similar to [14], a study [15] also proposed a task scheduling scheme based on the Storm topology, resources of worker nodes, and network traffic. In this study, the amount of communication between worker nodes is calculated by monitoring and plotted as a graph. The graph generated based on the amount of communication between worker nodes is converted into a spanning tree to perform tree partitioning using the information on the resources of worker nodes. A scheduling scheme for allocating tasks based on the information on the resources of worker nodes and partitioning is proposed.

A study [16] proposed mechanisms to dynamically enact the rescheduling and migration of tasks in a streaming dataflow from one set of virtual machines to another reliably and rapidly. They proposed two task migration strategies such as Drain-Checkpoint-Restore(DCR) and Capture-Checkpoint-Resume(CCR) in Storm by using Redis that is a distributed key/value store. The migration strategies allow a running streaming dataflow to migrate without any loss of in-flight messages or their internal tasks states, while reducing the time to recover and stabilize.

A study [17] proposed a dynamic scheduling algorithm to maximize the throughput of a heterogeneous distributed cluster environment. It considers cluster characteristics and the topology job and node attributes such as the computational complexities, data size, node processing powers, and link transfer bandwidths. They utilize the dynamic programming technique to identify and minimize the potential computational or communicational bottlenecks.

A study [18] proposed a thread-level non-stop task migration scheme called N-Storm that performs online task migrations in Storm without killing existing Executors or starting new Executors. N-Storm adds a key/value store on each worker node to make works be aware of the changes during task scheduling. Each worker in the same worker node can communicate with the supervisor through the key/value store. N-Storm performs

thread-level task migrations that improve the performance of task migration. Furthermore, they proposed several optimization schemes of N-Storm to get efficiency for multiple tasks migrations.

A study [19] modeled the elasticity problem for data stream processing as an integer linear programming problem, which is used to optimize different quality of service metrics such as the response time, the tuple processing time, the tuple transmission time, and the application downtime. They proposed a general formulation of the reconfiguration costs caused by operator migration and scaling in terms of application downtime. It implemented the proposed mechanisms on Apache Storm.

A study [20] proposed a dynamic scheduler that can redistribute the migrated tasks in a fair and fast way based on their previous work [21]. They perform the dynamic scheduling by estimating the load of the nodes to handle task migrations when the system parameters change such as the number of tasks and configuration of executors or nodes. It treats all the required operations (tuple transfer, tuple processing, and tuple packing) in a pipeline fashion based on their previous study [21] in order to reduce the overall processing time.

Studies [12,14–21] proposed task scheduling schemes based on various resources of worker nodes. T-Storm performs scheduling only based on traffic information, and thus, cannot solve the problem of load imbalance in worker nodes; it also entails delays in certain tasks if dynamic scheduling is not performed, as in [12]. Therefore, a task redistribution policy is needed based on real-time loads. Loads can be defined in various ways depending on the task type and the free resources of worker nodes. Task redistribution can be efficiently conducted if the policy varies with the load type. Finally, task deadlines may be differently defined based on the stream input time and type. Therefore, a measure in which task deadlines are also considered in dynamic scheduling is needed.

## 3. The Proposed Dynamic Task Scheduling Scheme

### 3.1. Characteristics e

Conventional schemes have the problem of delay in real-time processing since the redistribution of tasks and increased processing costs of worker nodes due to the computational complexities of the tasks and varying performances of worker nodes in a distributed environment are not considered. The states of worker nodes, task information, and processing costs need to be measured periodically to resolve such a problem. A dynamic task scheduling scheme is required in which the cause of loads on worker nodes and task priorities must be determined based on their information and tasks are redistributed from loaded nodes to free nodes based on the cause of loads.

In this paper, we propose a dynamic task scheduling scheme in which real-time loads on worker nodes in the Storm environment are considered. The proposed scheme distinguishes the loads on worker nodes based on their usage and throughput of the CPU, free memory size, and network load, following which the tasks are redistributed based on the load to balance the loads on worker nodes. Hence, the stream data are processed in real time to prevent bottlenecks. Figure 1 shows the system architecture of the proposed scheme, which consists of a Nimbus node, which plays the role of the master in the Storm environment, worker nodes, and a zookeeper node, which stores the state information of the worker nodes. The Nimbus node consists of a load table, which stores the information of the state of the worker nodes and the tasks processed in the worker nodes, and a scheduler, which proceeds with scheduling based on the load table. The worker nodes consist of executors that perform tasks and an NMonitor, which inspects the loads on the worker nodes. The workers execute an actual spout or bolt within a thread slot. NMonitor periodically transmits the task information to the zookeeper node. The zookeeper node creates the state information of a respective worker node based on the task information. The load table of the Nimbus node uses the zookeeper information to examine the load state of the worker nodes and redistributes the tasks. In Figure 1, the tasks are concentrated

on worker node 1, and the last input task, E, is redistributed to worker node 2 for balancing the loads.

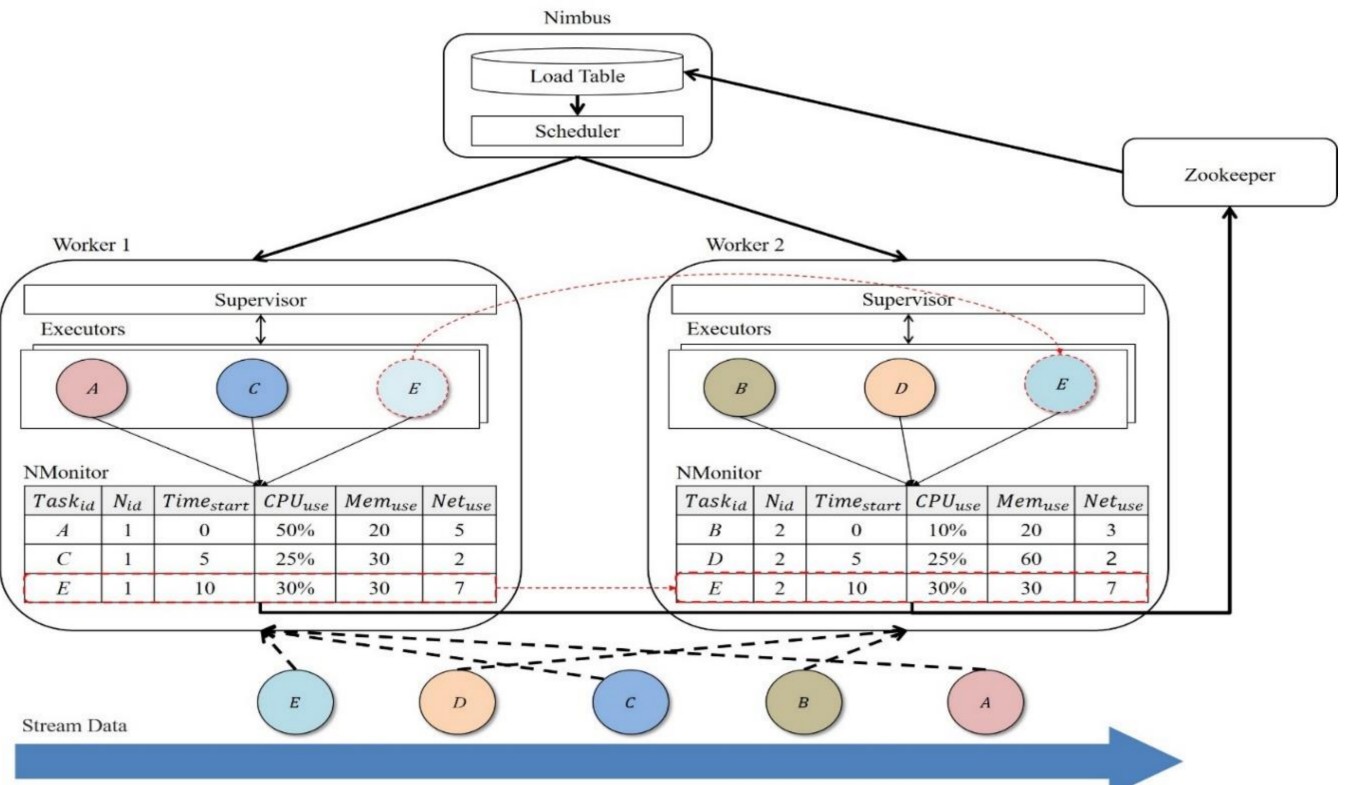

**Figure 1.** Overall architecture of the proposed local event detection scheme.

Figure 2 illustrates the overall processing procedure of the proposed scheme. When stream data are input, Nimbus, which is the master node, checks the loads using a load management module. Static scheduling is performed if the load is below a threshold, whereas dynamic scheduling is performed if the load exceeds the threshold. Static task scheduling uses R-Storm, developed in a previous study [12]. Dynamic scheduling is performed based on the load types, which are memory load, CPU load, and CPU/memory load. Dynamic scheduling redistributes the tasks based on the deadlines of the tasks being processed and allocates the newly input tasks to suitable worker nodes. Each worker node collects the information of the state and the tasks as well as the task throughput in real time. The data are collected as the NMonitor of each worker node sends task information and processing costs to the zookeeper node. Subsequently, the zookeeper node updates the load table of Nimbus every time the load state of a worker node changes based on the delivered information. Nimbus consistently performs static and dynamic scheduling based on the load table.

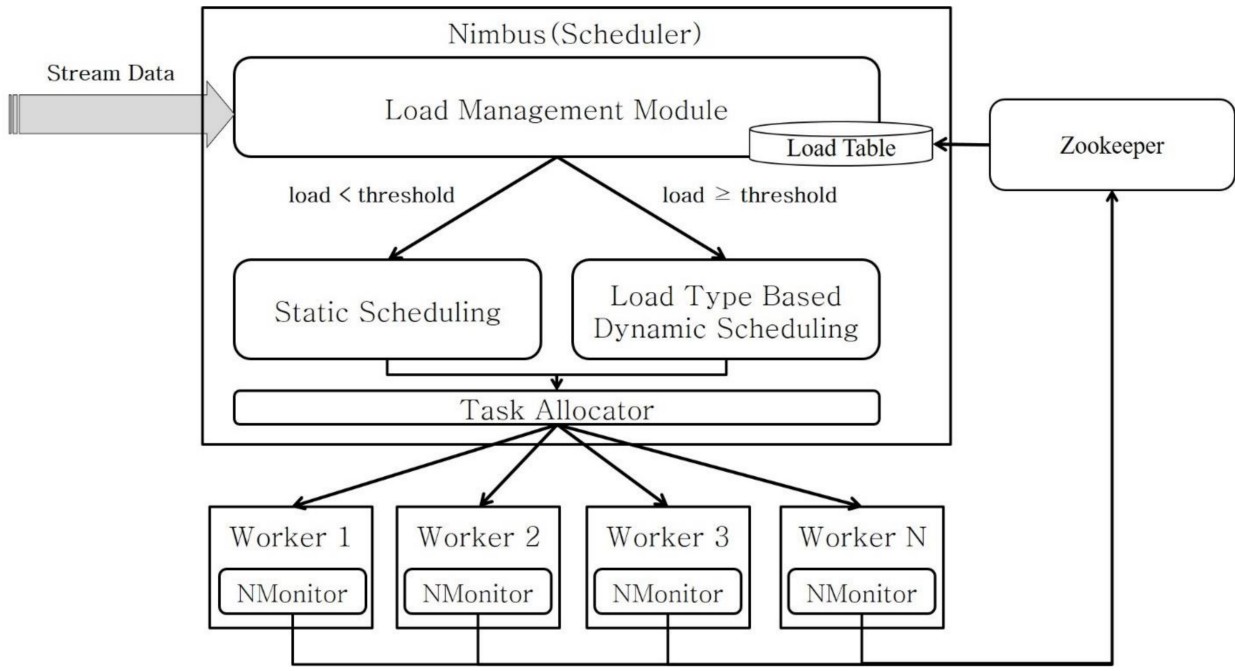

**Figure 2.** Scheduling process.

### 3.2. Collection and Management of Loads

Collecting and managing the load information of each worker node while processing real-time stream data in a distributed environment are crucial for identifying problems and load type. These processes are also used for identifying and solving various problems such as the CPU usage increase when there is a delay in a task or a task loss due to an increase in the memory usage. Furthermore, managing the loads on worker nodes and processing the costs of tasks, or the task information, are equally important as collecting loads on the worker nodes. This is because the loads on worker nodes become the selection criteria when the tasks are redistributed in a task scheduler. The loads on worker nodes are balanced as the task schedule redistributes the tasks based on the states of worker nodes and task information by collecting and managing the loads.

Table 1 summarizes the load table managed by Nimbus. The load table manages the CPU, memory, and network load information based on the tasks being processed by the worker nodes. $Node_{id}$ is an identifier of a worker node. $CPU_{use}$ and $CPU_{avg\text{-}th}$ represent the current CPU usage and the average throughput of the corresponding worker node. $Memory_{total}$, $Memory_{free}$, and $Network_{load}$ are the total memory size, free memory size, and network load, respectively. The load table is updated at every pre-determined unit (1 s). The CPU usage increases when the computation complexity of the task is high, whereas the memory usage increases when an excessive amount of stream data is input. Accordingly, loads can be examined based on the state of a worker node on which dynamic scheduling can be performed based on the load type.

**Table 1.** Load table information.

| Notation | Definition |
|---|---|
| $Node_{id}$ | ID of worker node |
| $CPU_{use}$ | CPU usage of worker node |
| $CPU_{avg\text{-}th}$ | Average CPU throughput of worker node |
| $Memory_{total}$ | Total memory size of worker node |
| $Memory_{free}$ | Free memory size of worker node |
| $Network_{load}$ | Network load |

Each worker node manages their own CPU usage, throughput, average throughput, and network load. Table 2 presents the information of the NMonitor managed by each worker node. NMonitor maintains the resource usage of the worker nodes used in each task. $Task_{id}$ and $N_{id}$ represent the identifiers of each task and worker node, respectively. $Time_{start}$ represents the starting time of a task and $Time_{dline}$ represents the deadline of the task, such that the task must be completed within *d* seconds from the starting time. NMonitor additionally records the resource usage of each task. For resource usage, $CPU_{use}$, $Memory_{use}$, and $Network_{use}$, which are the usages of the CPU, memory, and network are stored. When dynamic scheduling is required, a task to be migrated to a different worker node can be selected based on such information. Similar to the load table, the relevant information is managed by the zookeeper at a particular period to ensure Nimbus can examine the state of a worker node.

**Table 2.** NMonitor table information.

| Notation | Definition |
|---|---|
| $Task_{id}$ | Task ID |
| $Node_{id}$ | Worker node ID |
| $Time_{start}$ | Task input time |
| $Time_{dline}$ | Task deadline |
| $CPU_{use}$ | Task's CPU usage |
| $Memory^{use}$ | Task's memory usage size |
| $Network^{use}$ | Task's network usage |

### 3.3. Dynamic Task Scheduling

Based on the load information collected previously, the state of a worker node and the task information are used to determine the load type of the worker node. Figure 3 shows the decision tree of task scheduling. *n* indicates a worker node where a load is determined to be present since the CPU and memory usage exceed the threshold value ($\theta$). Task scheduling is performed based on the load type by checking the state of each worker node. First, the load table of Nimbus is examined, and the lowest memory usage of the worker nodes is determined. When the lowest value exceeds the threshold, the memory usage of all worker nodes is higher than the threshold, which is subsequently determined as the memory load, and memory load-based scheduling is performed accordingly. Alternatively, it is examined whether the CPU and memory usage of a particular worker node exceed the threshold. If a load occurs on at least one worker node, it is determined as the CPU and memory load, and CPU and memory load-based scheduling is performed. Finally, if the CPU usage of a particular worker node exceeds the threshold, it is determined as the CPU load, and CPU load-based scheduling is performed. When all three load types are not satisfied, R-Storm-based static scheduling is performed.

#### 3.3.1. Memory Load-Based Scheduling

When there is no free memory, it indicates that the memory of all worker nodes is full. In this case, a loss occurs even when a task is distributed to a worker node, and the task results cannot be used. Therefore, this paper proposes a method for delaying tasks with lower priorities based on the deadline and the CPU usage according to a priority queue.

Equation (1) is used for calculating the deadline of a task. $T_{n,cpu}$ denotes the CPU usage of task *T* in worker node *n*. $T_{n,start}$ denotes the starting time of task *T* in worker node *n*, whereas $T_{n,dline}$ represents the deadline of task *T* in worker node *n*. *Now* refers to the current time. The current time is divided by the sum of the starting time and deadline. At this time, the closer the current time is to deadline, the closer the value is to 1. Since the closer the value is to 1, the closer the task is to deadline, it should be processed as soon as possible. Therefore, tasks with high values assign high priorities. If a task fails to process within the deadline, it is calculated as higher than 1 because the start time plus deadline is less than the current time value. In this case, since the value is also greater than 1, even

tasks that exceed deadline are assigned high priority. A high weight is assigned to a task closer to the deadline considering the CPU usage and the deadline of the task. The value is high as the CPU usage is high and the deadline is close. $\alpha$ is a weight parameter that means how much more the task deadline will be considered.

$$T_{n,dead} = T_{n,cpu} * \left( \alpha * \frac{Now}{T_{n,start} + T_{n,dline}} \right) \tag{1}$$

When a memory load occurs, Equation (1) is used to calculate the priorities of the tasks. Subsequently, each worker node enqueues the tasks with the lowest priorities in the priority queue. This process is repeated until a memory space is in the acceptable space, and the tasks with higher priorities are processed first. When a task is processed in each worker node, those remaining in the priority queue are dequeued to continue the processing of the tasks. The memory load is reduced by this scheduling scheme; when a new task is requested during dynamic task scheduling based on the memory load, the task is enqueued based on the priority queue to prevent the memory load. For the CPU usage of a new task, the average CPU usage of the tasks in the priority queue is used.

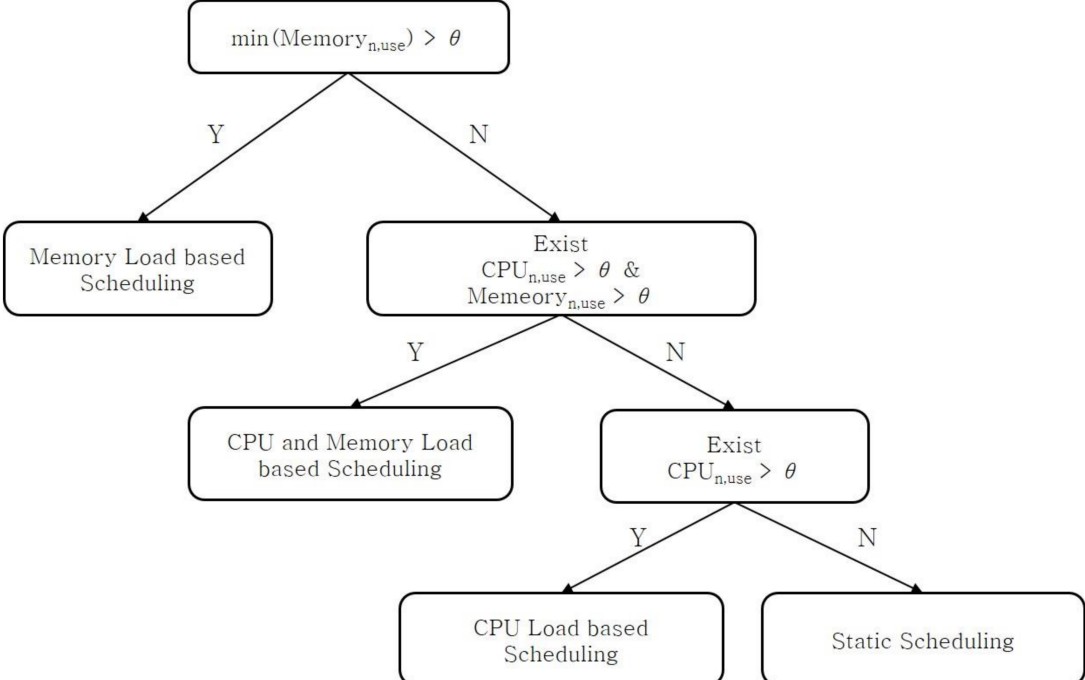

**Figure 3.** Task scheduling decision tree.

Figure 4 shows an example the memory load-based scheduling. Each worker node is allocated three tasks, where the CPU and memory usage of each task are assumed to be identical. Moreover, it is assumed that the memory usage of all worker nodes is ≥80% (no available memory). Assuming the current time (*now*) is 4, the starting time (*s*) and deadline (*d*) of each task are defined in advance. When the CPU usage and deadline of each task are calculated using Equation (1), they are provided at the top of the tasks. For instance, the starting time of *Task₁* is 1 and deadline is 5; thus, the result of Equation (1) is 0.66. In this study, a higher value indicates a higher priority. When a memory load occurs in this scenario, each worker node selects the tasks with the lowest priorities until free memory becomes available. Worker nodes 1, 2, and 3 select Tasks 4, 7, and 3, respectively. The selected task stops the task in a worker node and subsequently stores the respective task in a separately generated priority queue based on priorities. When *Task₁₀* is newly input, the priority of the new task is also calculated, and the task is loaded in the priority queue. The CPU usage of the new task is assigned the average CPU usage of the other

tasks because no information on the CPU usage is available. When a task is completed and the free memory is ready to accept a new task, each worker node loads a task on the priority queue and processes the tasks based on priorities. A loss of tasks is prevented by reducing the memory loads of the entire worker nodes.

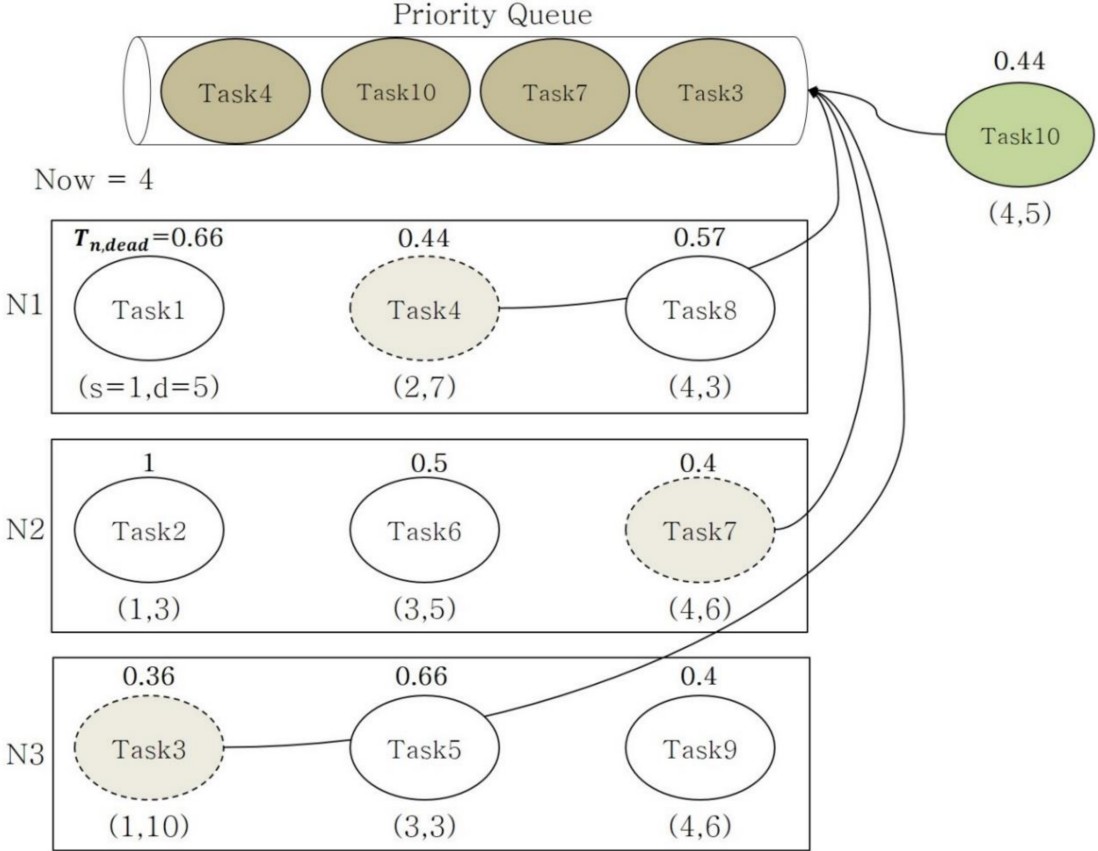

**Figure 4.** Memory load-based scheduling.

### 3.3.2. CPU Load-Based Scheduling

When the computation of a task being processed in a worker node is complex, computation of the CPU increases, thus increasing the CPU usage. In this case, a delay in the task processing occurs when a load is applied on the CPU. When the CPU usage increases owing to the computation complexity of a task, the loads on the worker nodes are classified based on the CPU usage to examine whether the task can be redistributed to free nodes with less loads. When redistribution is possible, the throughput from redistributing the respective task to a free node is calculated, and the task is redistributed if the free node does not become loaded. If it is determined that a load occurs when a task is redistributed to a free node, another free node is selected again to be processed.

Equation (2) is used for measuring the average CPU usage of each worker node, where $CPU_{exp}$ is the average CPU usage of the entire worker node, $N$ is the total number of worker nodes, and $CPU_{n,use}$ is the CPU usage of worker node $n$. The average CPU usage of all worker nodes is computed based on the above $CPU_{exp}$ values. Using the $CPU_{exp}$ values measured using Equation (2), the average CPU usage of all worker nodes is calculated, where the loaded and free nodes are distinguished based on the deviation between the $CPU_{exp}$ and CPU usage of each worker node.

$$CPU_{exp} = \frac{1}{N} \sum_{n=1}^{N} CPU_{n,use} \tag{2}$$

The loaded and free nodes can be determined using Equation (3) based on the CPU usage. Equation (3) finds the deviation in the CPU usage between $CPU_{exp}$ and the CPU usage of each worker node, where $CPU_{n,dev}$ is the deviation between $CPU_{exp}$ and the CPU usage of worker node $n$. A deviation value closer to a negative number indicates that the CPU usage exceeds the average value. A worker node with the smallest deviation value is classified as a loaded node, whereas that with the largest deviation is classified as a free node. When the free memory size of a worker node classified as a free node is less than the threshold, a loss of task may occur. In this case, a worker node with a large deviation is classified as a free node, excluding the worker node classified as a free node.

$$CPU_{n,dev} = CPU_{exp} - CPU_{n,use} \tag{3}$$

Among the loaded nodes classified using Equation (3), certain tasks being processed are migrated to the free nodes to balance the CPU usage of the loaded nodes. The tasks being processed in loaded nodes are selected based on the deadline and average CPU throughput of a worker node. If the average CPU throughput of a loaded node is higher than that of a free node, a task with a deadline far from the current time is selected, or a task with a close deadline is selected, to be migrated. The deadline value is calculated using Equation (1).

Figure 5 shows the occurrence of a load on worker node 3 caused by an increase in the CPU usage. In the following scenario, the average CPU usage (56%) is calculated using Equation (2), and the CPU deviation is calculated using Equation (3) to select worker node 1 with a high deviation value as a free node. In worker node 3, a task to be migrated is selected based on the deadline and the average CPU throughput. Because worker node 3 has a higher throughput than worker node 1, it is more efficient to process a task with a closer deadline from worker node 3. Therefore, $Task_{10}$ with a deadline that is relatively farther from worker node 3 is selected to be migrated. The selected task is migrated to worker node 1, and the NMonitor and load table are updated based on the migrated task. The average CPU usage is increased from 56% to 65%, whereas the load is balanced evenly.

### 3.3.3. CPU and Memory Load-Based Scheduling

A delay in tasks or loss may occur when a CPU and memory load occurs on one worker node. In this case, a node with a CPU and memory load and a free node are distinguished, and a task is selected to be migrated from the loaded node based on the CPU usage and the deadline and is redistributed to a free node. In this study, a memory and CPU load are defined as when a user uses the resources beyond the threshold value.

Equation (4) is used for selecting a task in a loaded node for redistribution of tasks. $T_{n,dead}$ is calculated using Equation (1). Specifically, the task with the highest CPU usage and deadline value in a loaded node is migrated. The selected task ($ST_i$) is migrated to a free node with a relatively smaller load to balance the loads.

$$ST_n = argmax(T_{n,dead}), \tag{4}$$

Similar to memory load-based scheduling, a task to be migrated from a loaded node is selected using Equation (4). It must be examined whether a delay occurs from the loads on a free node when the selected task that is migrated to a free node is being processed. Equation (5) is utilized for calculating the cost of migrating the task selected using Equation (4), where $ST_{n,network}$ is the network load of task $ST$ selected in worker node $n$, whereas $ST_{n,size}$ is the tuple size of task $ST$ selected in worker node $n$. The transmission cost of the task is calculated using the network load and the tuple size.

$$ST_{n,lat} = ST_{n,network} * ST_{n,size} \tag{5}$$

Equation (6) is used for calculating the expected processing costs, where $n$, $m$, and $ST$ are loaded worker node $n$, worker node m selected by migrating the task, and task $ST$

selected in Equation (4). $CPU_{m,avg\text{-}th}$ is the average CPU throughput of worker node $m$. Specifically, the sum of the expected throughput of task $ST$ migrated from worker node $n$ and the network transfer cost is the total cost of processing in worker node $m$. The task is migrated if this processing cost is lower than the cost of processing in worker node $n$, and the free memory of worker node $m$ can allow task $ST$.

$$\text{Cost}_{n,m,ST} = \frac{ST_{n,size}}{CPU_{m,avg-th}} + ST_{n,lat} \tag{6}$$

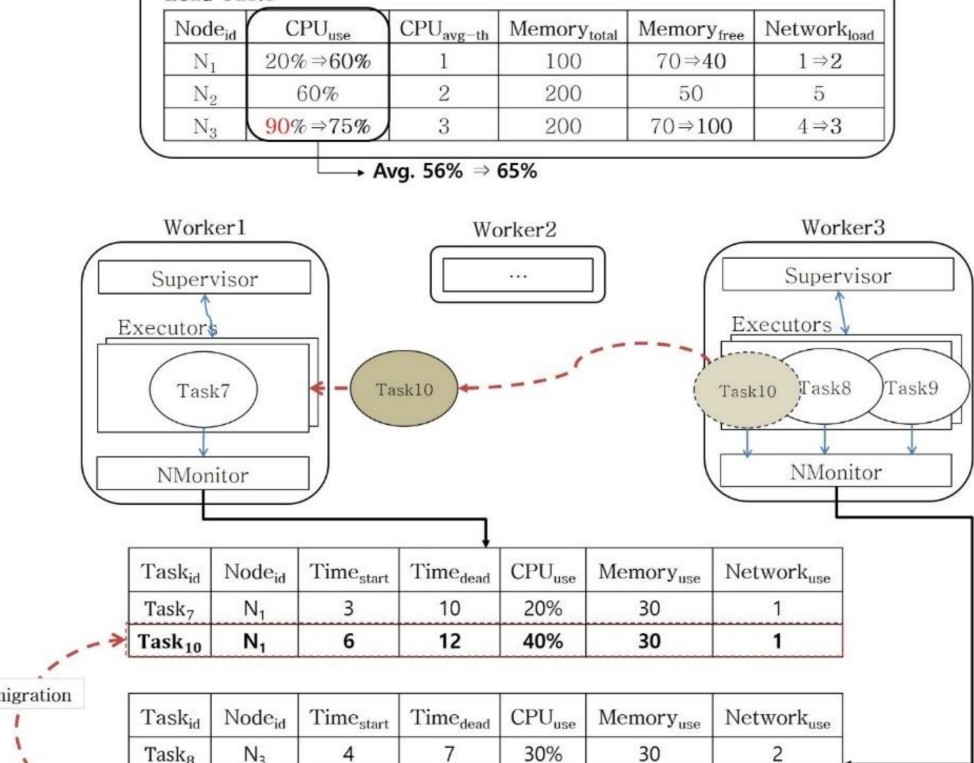

**Figure 5.** CPU load-based scheduling.

Figure 6 shows an example of CPU and memory load-based scheduling. In Nimbus, the CPU and memory of worker node 1 exceeded the threshold, and therefore, a load has occurred. The proposed scheme selected a task to be migrated using Equation (4). If the current time (*now*) is assumed to be 2, the deadline of $Task_3$ is closer at 33.33 (CPU usage 50 $\times$ task deadline 0.66), whereas the deadline of $Task_4$ is 30 (CPU usage 90 $\times$ task deadline 0.33) because the deadline is reasonably far from now even when the CPU usage is high. For the task $ST$ ($Task_3$) selected accordingly, the processing cost is calculated for each worker node using Equation (6). The calculation result shows that the average CPU throughput of worker node 3 is high, and thus, task $ST$ is migrated to worker node 3. NMonitor value of each worker node is updated after the migration in which the migrated $Task_3$ has decreased the CPU usage owing to the CPU performance of worker node 3; ultimately, the average CPU usage in the load table of Nimbus decreases from 91% to 78%.

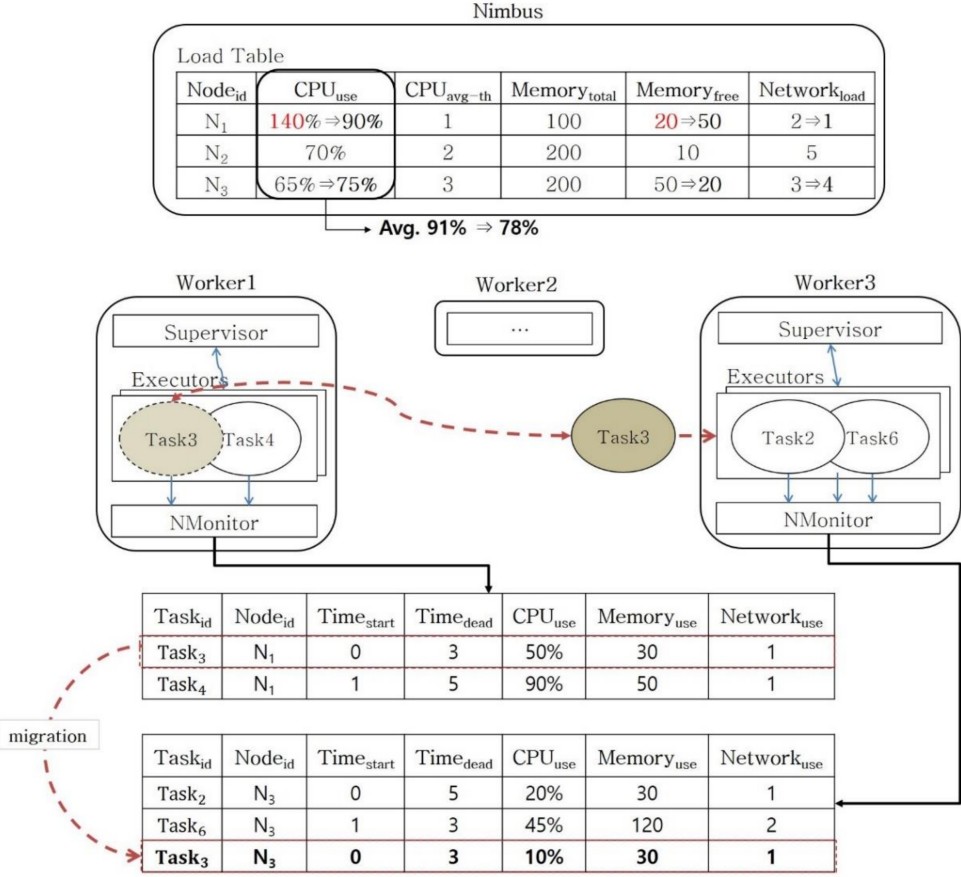

**Figure 6.** CPU and memory load-based scheduling.

## 4. Performance Evaluation

For verifying the excellence of the proposed dynamic task scheduling scheme, its processing speed was measured under each load condition, and the performance was evaluated for task scheduling proposed using R-Storm [12] and J. Fan [13]. The two existing methods we have chosen for performance evaluation are the most appropriate for performance comparison with the proposed method as they perform scheduling with only collectable memory usage, CPU, and network traffic information. The performance evaluation parameters are listed in Table 3. The environment consists of one master and ten worker nodes, in which the data processing speeds for complex and simple queries as well as the task delay of each worker node are evaluated. The threshold for determining a load is set as 80%.

**Table 3.** Performance evaluation parameters.

| Parameter | | Value |
|---|---|---|
| Operating System | | CentOS 6.6 |
| Master | Processor | Intel(R) Core(TM) i5-3570K |
| | Memory | 4 GB |
| Worker node 1 | Processor | Intel(R) Core(TM) i7-3610QM |
| | Memory | 4 GB |
| Worker node 2 | Processor | Intel(R) Core(TM) i5-3570K |
| | Memory | 2 GB |
| Worker node 3–10 | Processor | Intel(R) Core(TM) i5-3570K |
| | Memory | 2 GB |
| Disk | | 500 GB |
| Program Language | | Java |

Table 4 summarizes the types of data and queries used in the performance evaluation. The dataset used in the performance evaluation includes text files of 1 GB and 2 GB sizes consisting of English letters and of the same size consisting of random numbers. The performance evaluation data were transmitted in real time to construct a stream environment. Accordingly, the word count corresponding to a simple query and the k-means algorithm, which corresponds to a complex query, are processed [22]. The k-means algorithm is an algorithm for grouping data into $k$ number of clusters, where the distribution of the distances from each cluster is minimized. The factors that significantly influence the computational complexity of the k-means algorithm are Euclidean space $d$ and the number of clusters $k$. Even if the number of clusters is small, finding the optimal solution of the k-means algorithm in general Euclidean space $d$ is NP-hard [23,24]. Finding the optimal solution of $k$ number of clusters even in a low-dimensional Euclidean space is also NP-hard [25]. Therefore, the task complexity in the k-means algorithm is high because a heuristic technique is used. The processing speeds of the tasks and the real-time processing rates are comparatively analyzed by classifying the tasks into simple and complex tasks.

**Table 4.** Query type of performance evaluation.

| Parameter | | Value |
|---|---|---|
| Data size | | 1 GB, 2 GB |
| Data composition | | English alphabet, random number |
| Query | Simple Query | Word count |
| | Complex Query | k-means algorithm |

When processing real-time stream data, the task processing speed and the delay time vary with the load. Therefore, they must be examined with respect to the query of the proposed scheme. First, the speed of processing the tasks based on the worker node load and the rate of processing the tasks within the deadline are examined; subsequently, the performance is evaluated by comparing to previously proposed schemes. Text files of size 1 GB consisting of English alphabets and random numbers, respectively, are used.

Figure 7 illustrates the processing times of the proposed scheme for queries with respect to the CPU and memory usage. Figure 7a shows the processing time for a simple query, such as word count, and Figure 7b shows that for a complex query, such as k-means clustering. The processing times of the tasks varied based on the CPU usage, whereas no significant changes were observed with respect to the memory usage.

In addition, the quality of service (QoS) rate is examined for evaluating the performance. The QoS rate refers to the ration of the processed time to the deadline. For example, a QoS rate of 80% indicates that the task was completed within 80% of the deadline. This rate can be used to examine whether a task can be completed within a deadline. Figure 8 illustrates the QoS rates based on the query types. Figure 8a shows the QoS rate for a simple query. When the CPU usage is 20%, the tasks are processed within 90% of the deadline because the number of tasks is small. When the CPU usage is increased to 50–80% during real-time stream processing, the usage also increases but all the tasks are processed within the deadline. Figure 8b shows the QoS rate for a complex query. Similar to a simple query, the tasks are processed within 95% of the deadline if the usage is not high. Similarly, the proposed dynamic scheduling considering deadlines is valid because all tasks are processed within the deadline.

The proposed scheme is compared with existing scheduling schemes to verify the excellence of the former. The processing times are compared based on real-time data of various sizes and combination of queries (simple query + complex query). The conventional scheme, R-Storm, and the dynamic scheduling scheme proposed by Fan, adaptive scheduler, are comparatively evaluated.

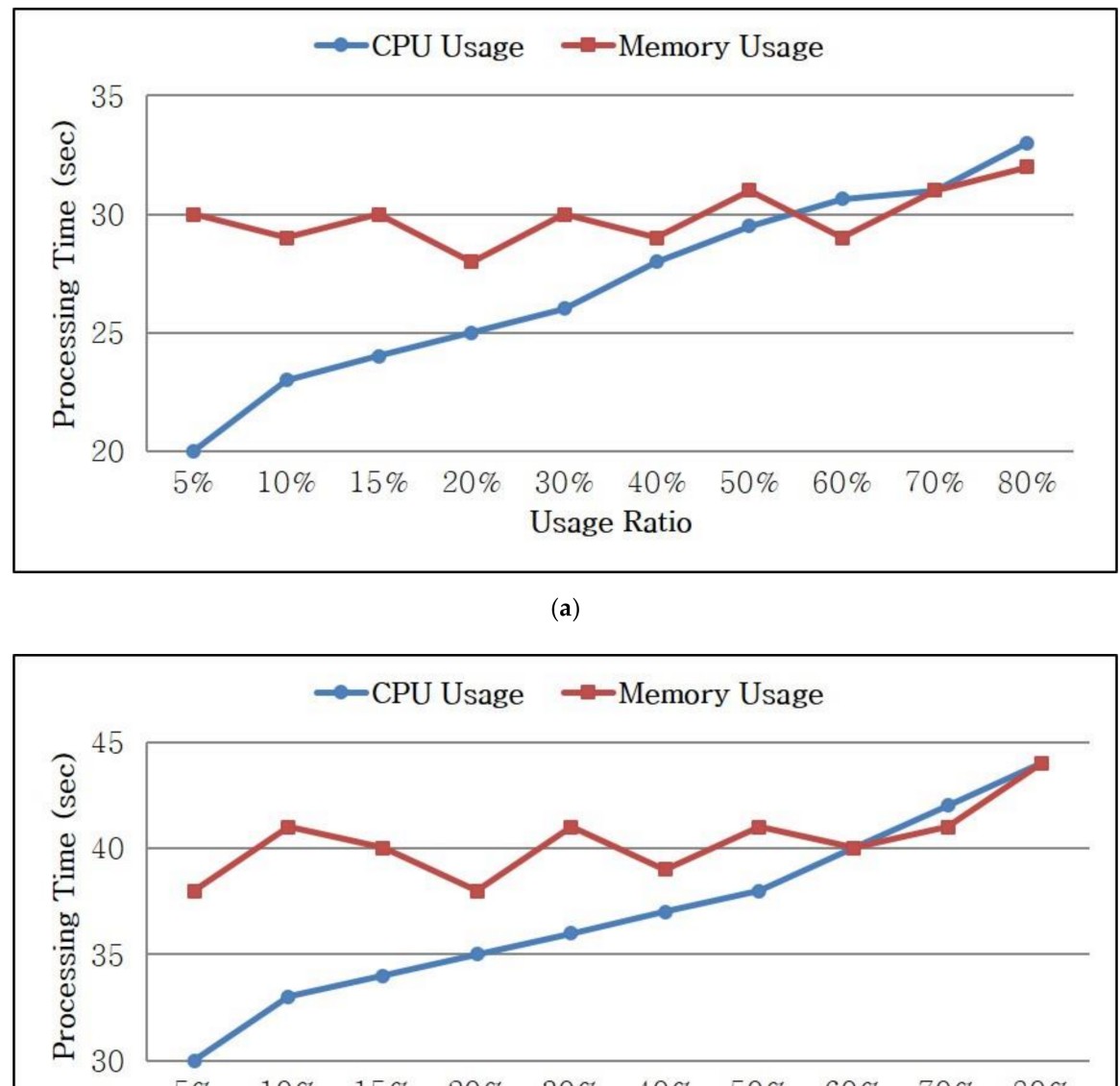

(**a**)

(**b**)

**Figure 7.** Processing times based on resource usage. (**a**) Processing time for simple query; (**b**) processing time for complex query.

Figure 9 shows the performance evaluation based on the data size and the query. Word count and k-means clustering were performed using text files of 1 GB and 2 GB consisting of English alphabets and random numbers separately. R-Storm has the highest task processing time and increases the processing costs of the worker nodes owing to its computational complexity. The adaptive scheduler scheme and the proposed dynamic scheduling scheme process real-time data within 45 s for the 2 GB data. The proposed scheme reduces the processing time by 11–15% on average by performing a scheduling in which the task deadline and various loads are considered.

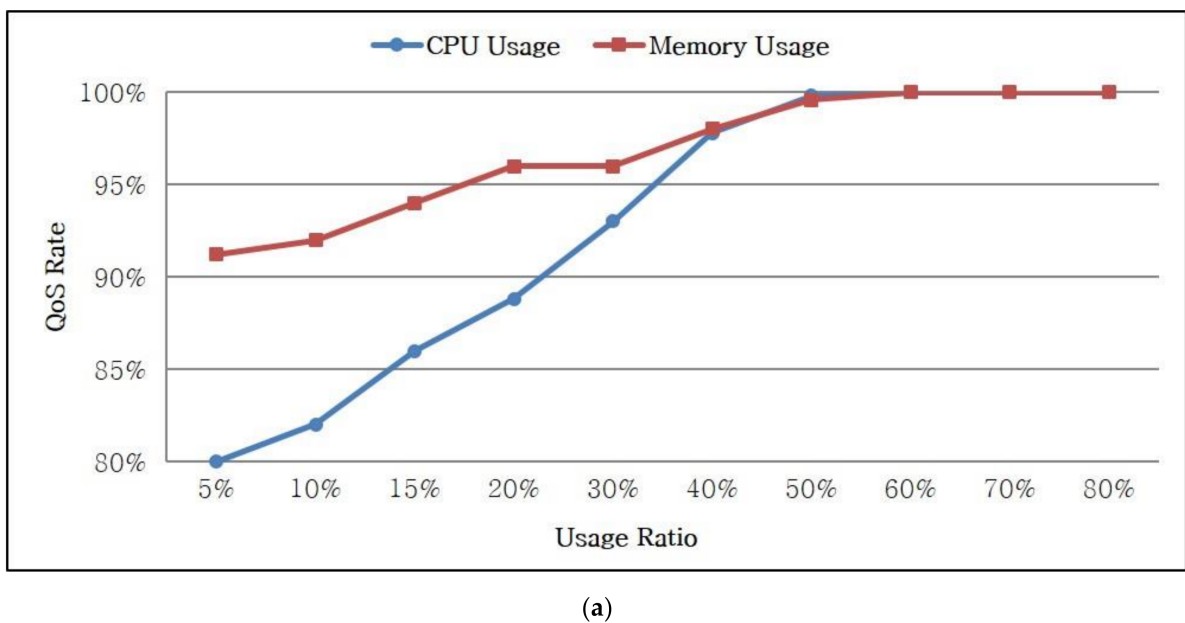

(**a**)

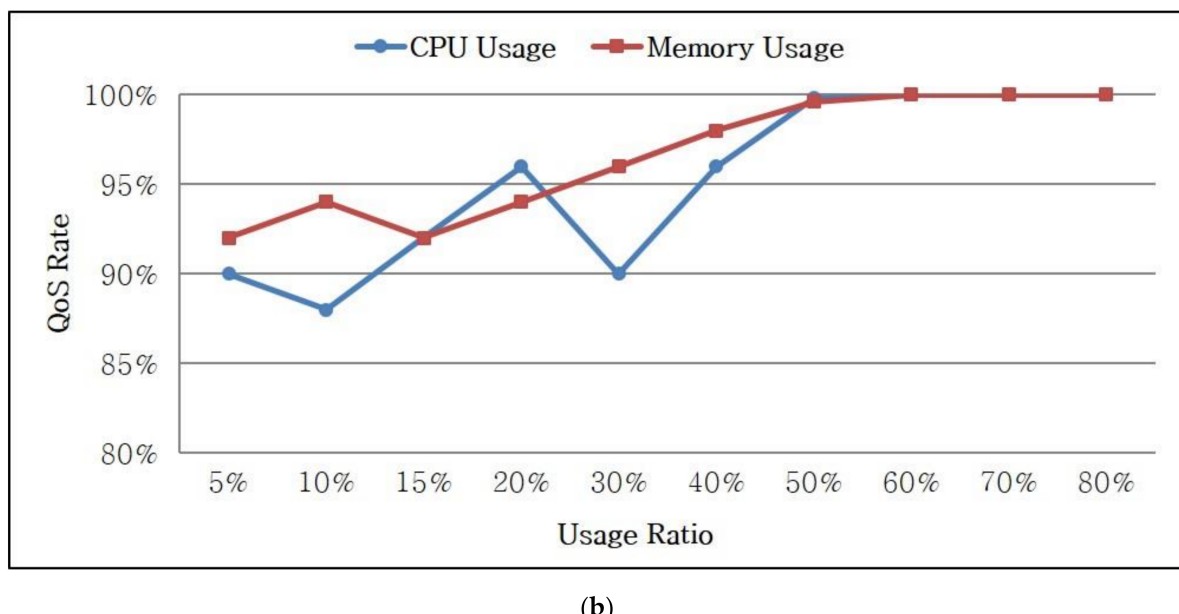

(**b**)

**Figure 8.** QoS rates based on resource usage. (**a**) QoS rate for simple query; (**b**) QoS rate for complex query.

In addition, the QoS rates of the tasks were examined. In Figure 10a, the average QoS rate of each worker node when processing a simple query is shown. The QoS rate is within 100% when the data size is small, thus indicating that all tasks are processed within the deadline. The proposed scheme has a relatively long delay time for a simple query. However, as shown in Figure 10b, the QoS rates of all conventional schemes exceed 100% for a complex query, thus having a delay in the tasks. This delay is caused by failing to consider the task deadline. The proposed scheme does not pass the deadline regardless of the data size because the deadline as well as CPU and memory usage of the worker nodes were considered, and exhibits a 9–11% improved performance compared to the existing schemes.

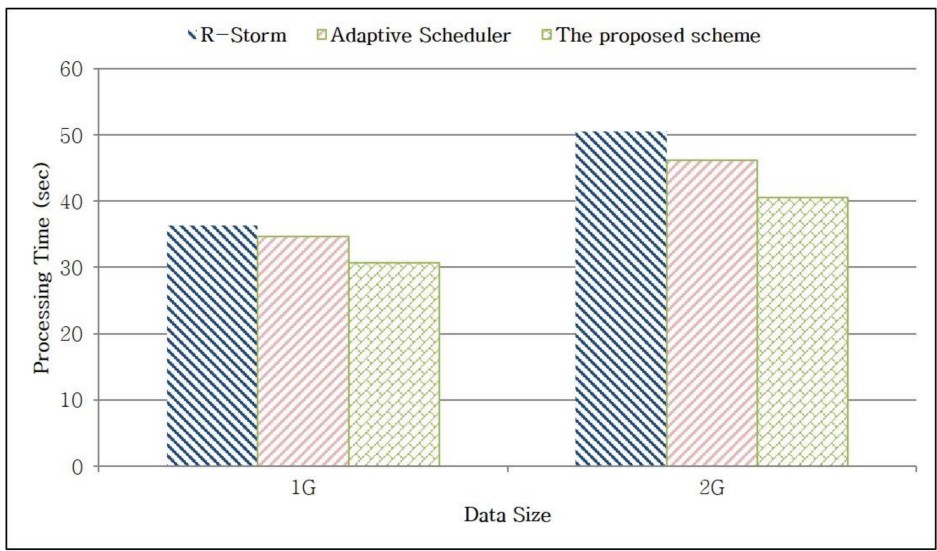

**Figure 9.** Query processing time based on data size.

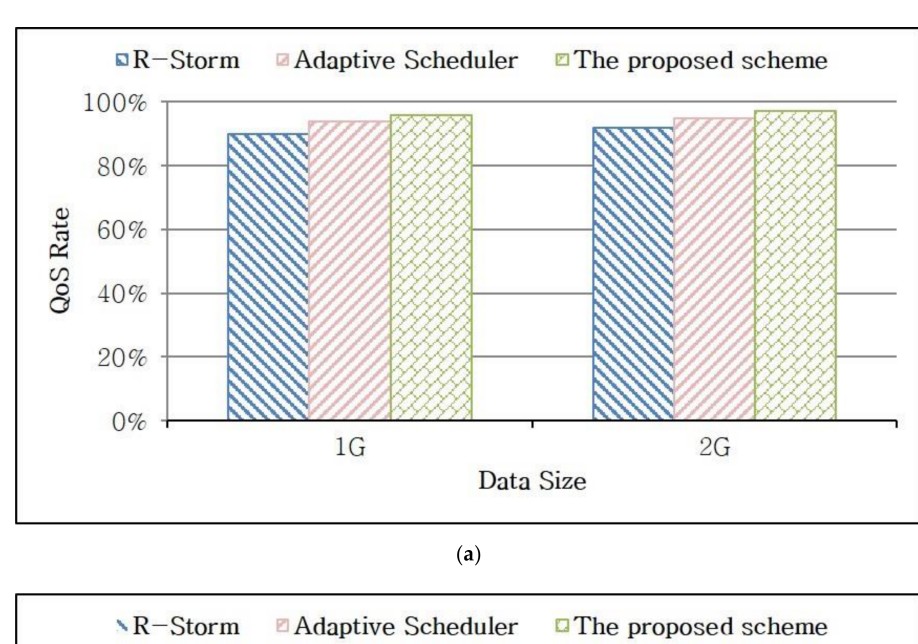

(**a**)

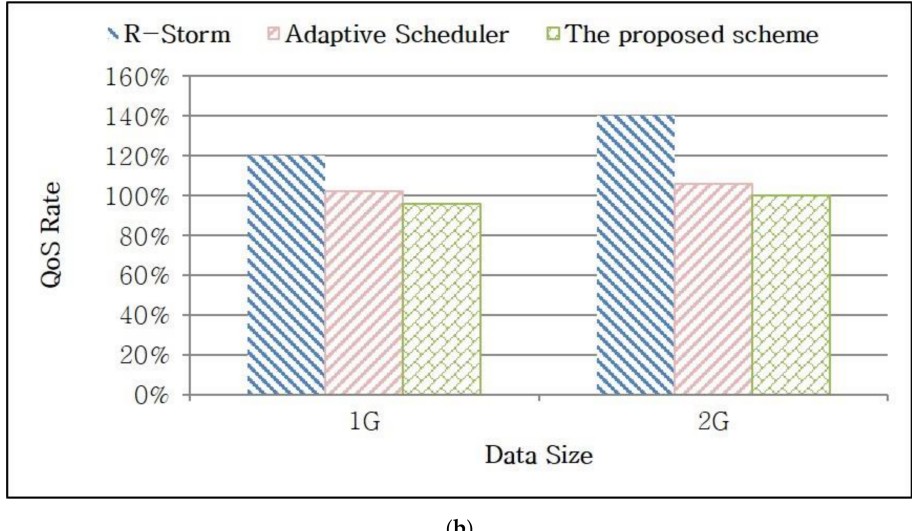

(**b**)

**Figure 10.** QoS rates based on data sizes. (**a**) QoS rate for simple query; (**b**) QoS rate for complex query.

## 5. Conclusions

In this paper, we have proposed a dynamic task scheduling scheme in which task deadline and node resources in Storm are considered. The proposed scheme considers the CPU usage, free memory size, and network load to calculate the load of each worker node and redistributes the tasks by assigning priorities based on the task deadline and the available resources. In this study, the loads were categorized into three types—CPU, memory, and CPU and memory loads—and dynamic task scheduling was performed with respect to the load type. Moreover, the throughput of each worker node was calculated by considering the state of a heterogeneous cluster, and the tasks with closer deadlines were distributed to the worker nodes with good performance to improve the QoS. The proposed scheme exhibited an approximately 15% more outstanding performance in terms of processing a complex query compared to existing schemes. Furthermore, both simple and complex queries were processed without passing the deadline in real-time processing. In future research, we will perform additional performance evaluations with the most recent existing dynamic task scheduling scheme. In addition, static task allocation and simple query processing performance will be improved by applying machine learning techniques.

**Author Contributions:** Conceptualization, D.C., H.J., J.L., K.B. and J.Y.; methodology, D.C., H.J., J.L., K.B. and J.Y.; validation, D.C., H.J., J.L. and K.B.; formal analysis, D.C., H.J., J.L. and K.B.; writing—original draft preparation, D.C., H.J., J.L. and K.B.; writing—review and editing, J.Y. All authors have read and agreed to the published version of the manuscript.

**Funding:** This work was supported by the National Research Foundation of Korea (NRF) grant funded by the Korean government (MSIT). (No. 2019R1A2C2084257), by the MSIT (Ministry of Science and ICT), Korea, under the Grand Information Technology Research Center support program (IITP-2021-2020-0-01462) supervised by the IITP (Institute for Information & communications Technology Planning & Evaluation), by "Cooperative Research Program for Agriculture Science and Technology Development (Project No. PJ01624701)" Rural Development Administration, Republic of Korea, and by Institute of Information & Communications Technology Planning & Evaluation (IITP) grant funded by the Korean government (MSIT) (No. 2014-3-00123, Development of High Performance Visual BigData Discovery Platform for Large-Scale Realtime Data Analysis).

**Institutional Review Board Statement:** Not applicable.

**Informed Consent Statement:** Not applicable.

**Data Availability Statement:** Not applicable.

**Conflicts of Interest:** The authors declare no conflict of interest.

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
