# Peer review of "Dynamic Task Scheduling Scheme for Processing Real-Time Stream Data in Storm Environments"

_applsci, doi:10.3390/app11177942_

Round 1

Reviewer 1 Report

This paper deals with the well-known problem of 
scheduling data streams during runtime. The paper is interesting, but there are certain aspects that the authors\
need to address. 

1) Related work: It is not complete. It only describes 5-6 papers, which are 
not adequate for a well-studied problem. Here are some more suggestions:

1. Shukla, A.; Simmhan, Y. Toward Reliable and Rapid Elasticity for Streaming Dataflows on Clouds. In Proceedings of the 2018
IEEE 38th International Conference on Distributed Computing Systems, Vienna, Austria, 2–6 July 2018; pp. 1096–1106

2. Souravlas, S.; Anastasiadou, S. Pipelined Dynamic Scheduling of Big Data Streams. Appl. Sci. 2020, 10, 4796

3. Al-Sinayyid, A.; Zhu, M. Job scheduler for streaming applications inheterogeneous distributed processing systems. J. Supercomput.
2020, 76, 9609–9628. 

4. Zhang, Z.; Jin, P.; Wang, X.; Liu, R.; Wan, S. N-Storm: Efficient Thread-Level Task Migration in Apache Storm. In Proceedings
of the IEEE 21st International Conference on High Performance Computing and Communications; IEEE 17th International
Conference on Smart City; IEEE 5th International Conference on Data Science and Systems (HPCC/SmartCity/DSS), Zhangjiajie,
China, 10–12 August 2019; pp. 1595–1602.

5. Cardellini, V.; Presti, F.; Nardelli, M.; Russo, G. Optimal operator deployment and replication for elastic distributed data stream
processing. Concurr. Comput. Pract. Exp. 2018, 30, e4334. [CrossRef]

6. Souravlas, S.; Anastasiadou, S.; Katsavounis, S. More on Pipelined Dynamic Scheduling of Big Data Streams. Appl. Sci. 2021, 11, 61. https://doi.org/10.3390/app11010061

The authors are strongly advised to add more papers.

Also: 
1.1) A simple presentation of techniques is trivial. Some type of classification of the strategies
as well as comparisons among them are required. A table that contains the main features (for example time complexity,
strategy being used) may be helpful

3. The proposed scheme 

3.3.1 The memory load based scheduler is difficult to understand. It is based on a single equation, but it \
is unclear what happens in case this equation produces equal priorities among the tasks. 
3.3.2 A dynamic scheduler should be directed towards saving the available memory. 
The authors claim that the memory load is reduced by this scheme but they do not
explain how this is done. Again, priorities play a major role, but it is 
unclear what happens when all the tasks are equally prioritized.  Also the
text starting at line 327

"The current time is divided by the sum of the starting time and deadline, and 328
a time closer to the deadline has a value closer to 1. If a task is not com- 329
pleted within the deadline, the weight of the deadline is higher than 1, 330
and thus, giving an even higher weight"

is rather unclear. Maybe, it should be rephrased. 

3.3.2 It is unclear what happens 

When the CPU usage increases owing to the computation complexity of a task, the loads on the worker nodes are classified based on the CPU 
usage to examine whether the task can be redistributed to free nodes with less loads. If so
the authors explain a process. If not? It is unclear.

3.3.3.
"In this case, a node with a CPU and 433
memory load and a free node are distinguished, and a task is selected 
to be migrated from the loaded node based on the CPU usage and the
deadline and is redistributed to a free node"

The authors select the "heaviest" task to migrate. 
The authors use Eq. 5 to compute the transmission cost and Eq. 6 to compute 
Eq. 6 to compute the processing cost. However, it is unclear how efficient this is.
For the moment, it is just a computation. Please, clarify the efficiency.

4 Results
The experimental results should be compared to more recent papers as well
R storm and Fan et al.'s are known, but rather old. The authors should 
justify this choice.

Please check the paper for grammatical errors.

Author Response

Dear Reviewer,

We would like to sincerely thank you for your attentive indications and good comments. Our paper is partially rewritten in order to revise and complement your comments. Please refer to the attached file about the detailed revisions.

Many thanks.

Jaesoo Yoo

Reviewer 2 Report

The authors address an interesting and up-to-date topic of dynamic scheduling based on a heterogeneous cluster consisting of various nodes having different performances. The Paper is very clear and consistent with several minor shortcomings that need to be improved.

Suggestions for improvement:

  • Section 2 “Related work” provides sufficient background to the introduction but it seems that lots of comments and references and repeated. Lines 52-74 are the essence of section 2. The authors are urged to consider merging these two sections into the single one.
  • Also, used references are relevant but consider adding additional ones. At the moment, both in sections 1 and 2 and later during the discussion the reference “Fan” pops out as the crucial one.

Overall, I believe that the article provides valuable content to the present body of knowledge but should reflect upon given comments.

Author Response

(The authors gave the same response as above.)

Round 2

Reviewer 1 Report

I believe that the authors have addressed most of my comments. The only week point is the experimental results. There should be more comparisons with newer schemes. This has been moved as future work. Apart from this weakness, the paper is suitable.

Author Response

Dear Reviewer,

We would like to sincerely thank you for your second attentive indications and good comments.  Please refer to the attached file about the detailed revisions.

Many thanks.

Best regards,

Jaesoo Yoo